# The Effect of Various Polishing Systems on the Surface Roughness of Two Resin Composites—An In Vitro Study

**Marta Ewa Szczepaniak [1,\*], Michał Krasowski [2] and Elżbieta Bołtacz-Rzepkowska [1]**

1   Department of Conservative Dentistry, Medical University of Łódź, 90-647 Łódź, Poland; elzbieta.boltacz-rzepkowska@umed.lodz.pl
2   University Laboratory of Materials Research, Medical University of Łódź, 90-647 Łódź, Poland; michal.krasowski@umed.lodz.pl
\*   Correspondence: marta.cwiklinska@stud.umed.lodz.pl; Tel.: +48-695-376-769

**Abstract:** The long-term success of a composite restoration largely depends on its smoothness, which can be achieved by the appropriate polishing tools and material selection. The purpose of this study was to evaluate the surface roughness of two composite materials after the application of selected polishing systems. Filtek Ultimate (FU) and Filtek Z250 (FZ) were tested. Forty specimens of each material were prepared. After polymerization under a Mylar strip, the surface roughness of five samples from each group was measured. Subsequently, all specimens were ground by 600 grit sandpaper. The surface roughness of five samples per group was tested again. The samples from each group were randomly assigned to eight subgroups, and polished by Sof-Lex, Sof-Lex Diamond Polishing System, Super Snap, One Gloss, Astrobrush, Stainbuster, Enamel Shiny, and Jiffy Polishing System. The collected data were analyzed using the Shapiro–Wilk and Kruskal–Wallis tests. The lowest Ra coefficient after polishing is found in the Super Snap groups (FU—0.077 μm, FZ—0.085 μm). The lowest Rlr coefficient is measured in the Enamel Shiny group for FU (1.000), and for Sof-Lex, Sof-Lex Diamond Polishing System, and Jiffy Polishing Kit for FZ (1.001), and only slightly higher for Super Snap (FU—1.001, FZ—1.002). The roughest-measured surface is in the One Gloss group for FU (Ra—0.657 μm, Rlr—1.009), and Astrobrush group for FZ (Ra—0.525 μm, Rlr—0.011). Additionally, it was not confirmed that the nanoparticle material (FU) demonstrates better results than the microhybrid one (FZ). Different polishing systems produce varying surface roughness. The most effective polishing system is Super Snap. The structure of composites does not significantly affect their surface roughness after polishing.

**Keywords:** finishing; polishing; surface roughness; composite resins

## 1. Introduction

Both researchers and clinicians agree that achieving optimal filling smoothness is a very important factor that enhances the longevity and esthetics of tooth restoration [1]. The appropriate smoothness of a restorative material limits the accumulation of dental plaque, [2], preventing the development of secondary caries and gingival inflammation [3]. Accurate polishing significantly improves the aesthetics of the filling [4]. The rough surface intensifies the absorption of chemical compounds from beverages and food by bacterial biofilm; it causes faster discoloration and filling degradation. Unevenness in tangible material results in discomfort, and excessive wear of the opposing teeth [5].

Material smoothness is measured through its roughness, i.e., optically recognizable or mechanically noticeable surface irregularities, not resulting from its shape, but the method of processing and the instruments used. Roughness, unlike surface waviness, is a concept that refers to unevenness, with relatively small vertex distances. The most commonly used parameter while assessing the smoothness of material surface is Ra (μm): the arithmetic mean of the vertical departures of the roughness profile from the mean line [6].

Researchers emphasize three clinically important limit Ra values—1 μm, 0.3 μm, and 0.2 μm. Many authors propose an Ra value of 1 μm as a threshold for visibly acceptable surface roughness [7] that ensures a natural gloss. Hachiya and colleagues propose that the only esthetically acceptable surface finish is a reflective one, and find that a surface is considered reflective when its imperfections are well below 1 μm [8].

Changes in surface roughness in the order of 0.3 μm are easily detected by the tip of the tongue [9], Thus, when the composite surface roughness exceeds 0.3 μm, a patient may report discomfort.

However, the most important and most restrictive Ra value is 0.2 μm. Studies show that when the roughness of material surface rises above 0.2 μm, the bacterial adhesion increases rapidly [10]. On the other hand, bacterial adhesion no longer decreases below this value, despite further polishing [11]. Not as common in the literature, but potentially clinically significant, is the Rlr coefficient. It is the ratio of the actual measured length of the roughness profile to the sampling length. Rlr indicates the degree of development of the roughness profile. The more developed the surface, the greater the potential area for bacterial adhesion, and a larger contact area of the filling with chemical compounds from beverages and food, and the intensity of their absorption. These factors may influence secondary caries formation, faster discoloration, and filling degradation.

A number of studies show that the smoothest surface is achieved when binding the material under a Mylar strip [12–14]. Another benefit of a Mylar strip polymerization is a lack of an oxygen inhibition layer. However, using a Mylar strip increases the c-factor, which could be unfavorable for the mechanical properties of restorations. In addition, the resulting superficial layer is resin-rich, and might contain some voids [8], therefore, the material polymerized without a Mylar strip or glycerin requires surface preparation. In addition, it is usually necessary to remove filling excess in order to properly imitate the tooth anatomy, and adjust the filling to the occlusion. As a result of these actions, the outer surface of the filling becomes rough, and should be further prepared.

In dentistry, optimal filling smoothness is obtained by performing finishing and polishing procedures. Finishing is the gross contouring of a restoration to obtain a desired shape [15]. The finishing procedure can remove the excess material with a particle size of more than 25 μm [16]. Polishing is carried out after finishing, in order to eliminate minor scratches on the surface of the filling and give it a smooth, reflective gloss [17]. In this procedure, particles lesser than 25 μm are removed [16]. Both finishing and polishing procedures are performed using dedicated tools. Their choice on the market is very wide. Despite numerous studies, no consensus has yet been reached as to the optimal and universal method of polishing the surfaces of composite fillings.

Final material roughness is a result of many factors, associated with both the material and the polishing system [18,19]. Factors that depend on the material itself include the filler load, its distribution in the matrix, particle size, and degree of filler hardness and matrix [9,16,20].

Polishing systems differ in base material and flexibility, hardness of the embedded abrasive particles, and the shape of applied instruments. It is possible that the matrix composition, and the manner in which the particles are bound within the matrix, affect polishing efficiency [5]. In addition, the operator's actions (speed and type of movements), tip rotation speed, the presence or absence of water cooling, polishing time, and the way of finishing, i.e., whether these procedures are carried out immediately after the material setting or are postponed, also have a considerable impact [21].

It can be assumed that there is no single factor that accurately predicts the effect of abrasives on restorative materials. Therefore, different polishing systems and procedures should be used, as well as various materials; this is confirmed by many studies [22,23]. Due to the large selection of polishing systems on the market, they should be evaluated to verify which of them yields the best polish effect on a particular composite. Thus, the purpose of this study was to evaluate the surface finish of two direct resin composites, one nanoparticle and one microhybrid, after the application of eight different polishing systems.

The null hypotheses of this study were:

There is no difference in surface roughness between the different polishing systems when used on the same composites.

There is no difference in surface roughness between the polished resin composites after the application of a specific polishing system.

## 2. Materials and Methods

Eighty cylindrical composite samples were prepared: 40 samples of Filtek Ultimate (FU), nanocomposite, and 40 of Filtek Z250 (FZ), microhybrid composite (both from 3M Oral Care, Alexandria, MN, USA). The composition of the materials is listed in Table 1. All specimens were of equal thickness (2 mm) and diameter (10 mm), in shade A2.

**Table 1.** Composition of the resin composites used in the study.

| Brand Name | Composite Type | Composition | Manufacturer |
|---|---|---|---|
| Filtek Ultimate | Nanocluster | Matrix: Bis-GMA, UDMA, TEGDMA, PEGDMA, Bis-EMA resins<br>Filler: non-agglomerated/non-aggregated 20 nm silica filler, non-agglomerated/non-aggregated 4 to 11 nm zirconia filler, aggregated zirconia/silica cluster filler (composed of 20 nm silica and 4 to 11 nm zirconia particles). Average cluster particles 0.6 to 10 μm; filler loading—55.6% volume. | 3M Oral Care, Alexandria, MN, USA |
| Filtek Z250 | Microhybrid | Matrix: Bis-GMA, UDMA, Bis-EMA resins<br>Filler: zirconia/silica, particle size range of 0.01 to 3.5 μm, average particle size—0.6 μm; filler loading—60% volume | 3M Oral Care, Alexandria, MN, USA |

A certain amount of material was applied to the silicone mold, covered on each side with a Mylar strip, and pressed with a smooth glass plate to remove excess. Samples were polymerized with a 1200 mW/cm$^2$ intensity lamp (The CURE TC-01, Spring Health Products, Norristown, PA, USA). The light intensity was measured and controlled by Light Meter 200 (Jovident Systems, Baarn, The Netherlands). The exposure time was adapted to the manufacturer's recommendations: 10 s for Filtek Ultimate, 20 s for Filtek Z250.

After 24 h storage in distilled water, the roughness of five randomly selected samples from each group was measured (positive control group—Mylar strip). Then the surface of all samples were ground with 600 grit sandpaper, in order to remove a weak resin-rich layer and obtain a standard finish surface before final polishing. The grinding of every sample was performed with water cooling by a single operator for five seconds. The obtained material roughness is adequate to a clinical situation after adjusting the restoration and before its high gloss polishing. After grinding, the roughness profile of five randomly selected samples from each group was tested again (negative control group—ground).

Samples in two main groups, FU and FZ, were randomly divided into eight subgroups (five samples for each), and polished using selected tools and polishing systems, in accordance with the manufacturers' recommendations. The list of applied polishers, their composition, and parameters used during application are included in Table 2. To reduce variability, all specimen preparation and polishing procedures were performed by the same operator.

After 24 h, the geometrical structure of the material surface was examined. The quantitative assessment was performed using a SJ-410 Surftest contact profilometer (Mitutoyo, Kawasaki, Japan), while estimating the roughness coefficients. The roughness of the samples was measured by traversing three tracings at three different locations, inclined to each other by an angle of 45 degrees on each specimen. The sampling length was 0.8 mm, and the number of intervals (N) was 5. The cutoff length was 2.5 μm, and the sampling speed was 0.2 mm/s. The measurements were carried out for two roughness parameters: Ra

and Rlr. Ra is the arithmetic mean of the absolute values of evaluation profile deviations from the mean line (ISO 1997). Ra is defined over the entire evaluation length. Rlr is the expansion length ratio. It is the ratio of the actual distance traveled by the profilometer needle to the sampling length (ISO 1997). This ratio describes the degree of depression in the evaluation profile.

**Table 2.** List, composition and parameters used during application of the polishing systems.

| Materials | Matrix | Abrasives | Water Spray | Speed (rpm) | Time (s) | Manufacturer |
|---|---|---|---|---|---|---|
| Sof-Lex | Thermoplastic elastomer | Aluminum oxide | - | 1. Coarse—10.000<br>2. Medium—10.000<br>3. Fine—30.000<br>4. Superfine—30.000 | 15<br>15<br>15<br>15 | (3M Oral Care, Alexandria, MN, USA) |
| Sof-Lex Diamond Polishing System | Thermoplastic elastomer | Aluminum oxide (1), diamond particles (2) | + | 1. Pre-polishing—20.000<br>2. High gloss polishing—20.000 | 30<br><br>30 | (3M Oral Care, Alexandria, MN, USA) |
| One Gloss | Silicone | Aluminum oxide | + | Heavy pressure polishing (~1.0 N)—5.000<br>Light pressure polishing (~0.3 N)—10.000 | 20<br><br>20 | (Shofu INC., Kyoto, Japan) |
| Super Snap | Polyester | Silicon carbide (1,2)<br>Aluminum oxide (3,4) | - | 1. Contouring—15.000<br>2. Finishing—15.000<br>3. Polishing—15.000<br>4. Super polishing—15.000 | 20<br>20<br>20<br><br>20 | (Shofu INC., Kyoto, Japan) |
| Astrobrush | Polyamide | Silicon carbide | - | 5.000 | 30 | (Ivoclar Vivadent AG, Schaan, Lichtenstein) |
| Stainbuster | Composite material, a resin reinforced by zircon-rich fiberglass | Zircon-rich fiberglass | + | 10.000 | 30 | (Abrasive Technology, Lewis Center, OH, USA) |
| Enamel Shiny | Goat hair brush (3,4)Felt wheel (5) | Diamond (1)<br>Diamond (2)<br>A—diamond<br>B—diamond<br>C—aluminium oxide | +/− | 1. Diamond bur<br>2. Rubber—5.000<br>3. Paste A—10.000<br>4. Paste B—10.000<br>5. Paste C—10.000 | 15<br>15<br>15<br>15<br>15 | (Micerium S.p.a., Avegno, Italy) |
| Jiffy Polishing System | Silicone | Aluminum oxide (1,2,3), Diamond, Silicon Carbide (HiShine) | + | Coarse—10.000<br>Medium—8.000<br>Fine—5.000<br>Superfine—3.000<br>Brush—3.000 | 15<br>15<br>15<br>15<br>15 | (Ultradent Products. INC., South Jordan, UT, USA |

The obtained results were subjected to statistical analysis using Statistica v. 12 software (StatSoft, Kraków, Poland). Means, medians, standard deviations (SD), minimum, and maximum values were calculated, and statistically significant differences were noted. The Shapiro–Wilk test found part of the data not to be distributed normally. Therefore, the mean Ra and Rlr values obtained for each material in individual research subgroups were compared using the non-parametric Kruskal–Wallis test. $\alpha = 0.05$ was used to indicate significance.

Pictures of individual samples were also taken at 100 times magnification using the Olympus optical microscope, model BX51.

## 3. Results

The mean values, standard deviations, and medians of surface roughness (Ra [μm], Rlr) for each resin composite are given in Tables 3–6.

**Table 3.** Descriptive statistics, Ra [μm] for Filtek Ultimate.

| Method | Mean | SD | Median |
|---|---|---|---|
| Mylar strip | 0.060 | 0.013 | 0.057 |
| Ground | 0.736 | 0.152 | 0.788 [A,B] |
| Sof-Lex | 0.092 | 0.025 | 0.095 [A,C] |
| Sof-Lex Diamond Polishing System | 0.415 | 0.054 | 0.419 |
| One Gloss | 0.670 | 0.094 | 0.657 [C,D] |
| Super Snap | 0.085 | 0.015 | 0.077 [B,D] |
| Astrobrush | 0.542 | 0.117 | 0.604 |
| Stainbuster | 0.458 | 0.094 | 0.470 |
| Enamel Shiny | 0.277 | 0.078 | 0.287 |
| Jiffy Polishing Kit | 0.224 | 0.032 | 0.232 |

A, B, C, D—statistically significant differences.

**Table 4.** Descriptive statistics, Ra [μm] for Filtek Z250.

| Method | Mean | SD | Median |
|---|---|---|---|
| Mylar strip | 0.051 | 0.019 | 0.050 |
| Ground | 0.757 | 0.228 | 0.625 [A,B] |
| Sof-Lex | 0.157 | 0.049 | 0.144 [A] |
| Sof-Lex Diamond Polishing System | 0.198 | 0.037 | 0.189 |
| One Gloss | 0.473 | 0.078 | 0.447 [C] |
| Super Snap | 0.089 | 0.011 | 0.085 [B,C,D] |
| Astrobrush | 0.531 | 0.069 | 0.525 [D] |
| Stainbuster | 0.173 | 0.036 | 0.169 |
| Enamel Shiny | 0.333 | 0.075 | 0.342 |
| Jiffy Polishing Kit | 0.235 | 0.037 | 0.222 |

A, B, C, D—statistically significant differences.

**Table 5.** Descriptive statistics, Rlr for Filtek Ultimate.

| Method | Mean | SD | Median |
|---|---|---|---|
| Mylar strip | 1.001 | 0.000 | 1.001 |
| Ground | 1.023 | 0.005 | 1.025 [A,B,C] |
| Sof-Lex | 1.001 | 0.000 | 1.001 [A] |
| Sof-Lex Diamond Polishing System | 1.002 | 0.000 | 1.002 |
| One Gloss | 1.009 | 0.003 | 1.009 [D] |
| Super Snap | 1.001 | 0.000 | 1.001 [B] |
| Astrobrush | 1.010 | 0.003 | 1.009 [E] |
| Stainbuster | 1.005 | 0.001 | 1.004 |
| Enamel Shiny | 1.000 | 0.000 | 1.000 [C,D,E] |
| Jiffy Polishing Kit | 1.001 | 0.001 | 1.001 |

A, B, C, D, E—statistically significant differences.

**Table 6.** Descriptive statistics, Rlr for Filtek Z250.

| Method | Mean | SD | Median |
|---|---|---|---|
| Mylar strip | 1.001 | 0.000 | 1.001 |
| Ground | 1.023 | 0.005 | 1.019 [A,B] |
| Sof-Lex | 1.001 | 0.000 | 1.001 [A] |
| Sof-Lex Diamond Polishing System | 1.001 | 0.000 | 1.001 [B] |
| One Gloss | 1.005 | 0.003 | 1.003 |
| Super Snap | 1.003 | 0.002 | 1.002 |
| Astrobrush | 1.009 | 0.004 | 1.011 |
| Stainbuster | 1.003 | 0.000 | 1.003 |
| Enamel Shiny | 1.002 | 0.001 | 1.002 |
| Jiffy Polishing Kit | 1.002 | 0.001 | 1.001 |

A, B—statistically significant differences.

The analysis of the roughness coefficient measurements (Ra, Rlr) shows that the surface of both materials polished with each of the tools or systems is rougher than in the positive control group (materials polymerized under a Mylar strip). The highest roughness coefficient values are obtained in the negative control group (after grinding with 600 grit sandpaper): Ra 0.788 μm, Rlr 1.025 for FU, and Ra 0.625 μm, Rlr 1.019 for FZ. Each of the applied tools and systems results in improved surface smoothness for both ground materials, as expressed in the lower values of the roughness parameters. The lowest Ra values are obtained in the group polished with Super Snap (for FU—0.077 μm, for FZ—0.085 μm) and Sof-Lex (for FU—0.095 μm, for FZ—0.144 μm) [Tables 3 and 4, Figure 1].

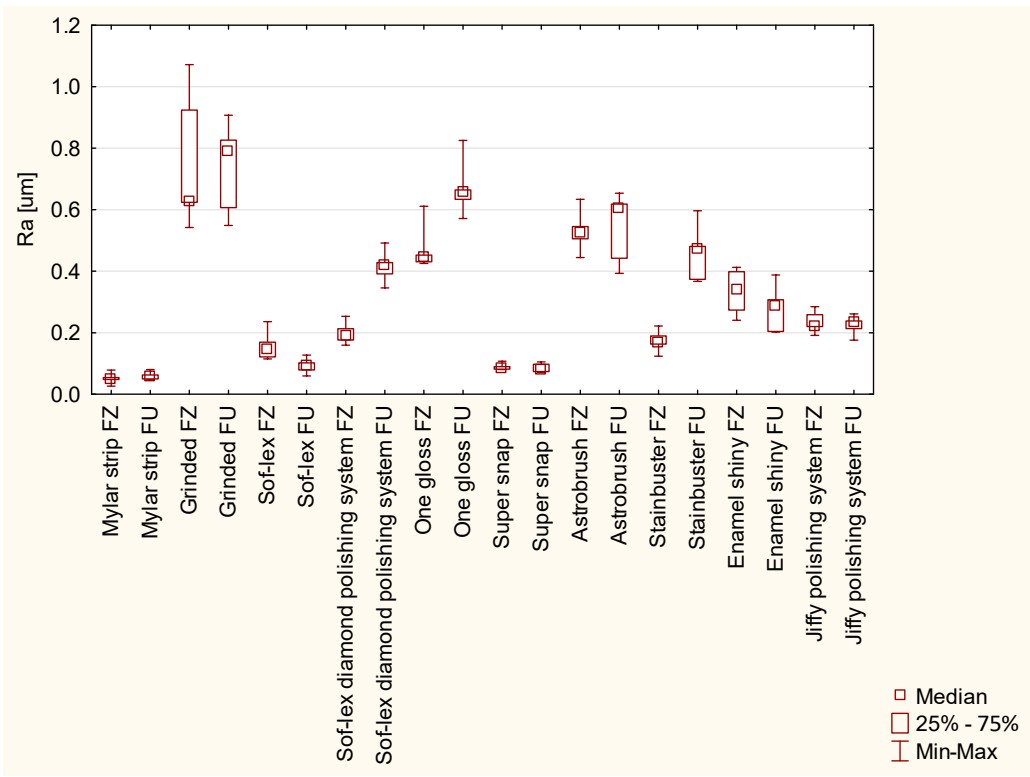

**Figure 1.** Median analysis, Ra comparison between FU and FZ.

The lowest Rlr values are obtained after polishing the samples with the Enamel Shiny for FU (1.000), and with the Sof-Lex, the Sof-Lex Diamond Polishing System, and Jiffy Polishing Kit for FZ (1.001) [Tables 5 and 6, Figure 2]. The value is only slightly higher after polishing both materials with Super Snap (1.001 for FU and 1.002 for FZ).

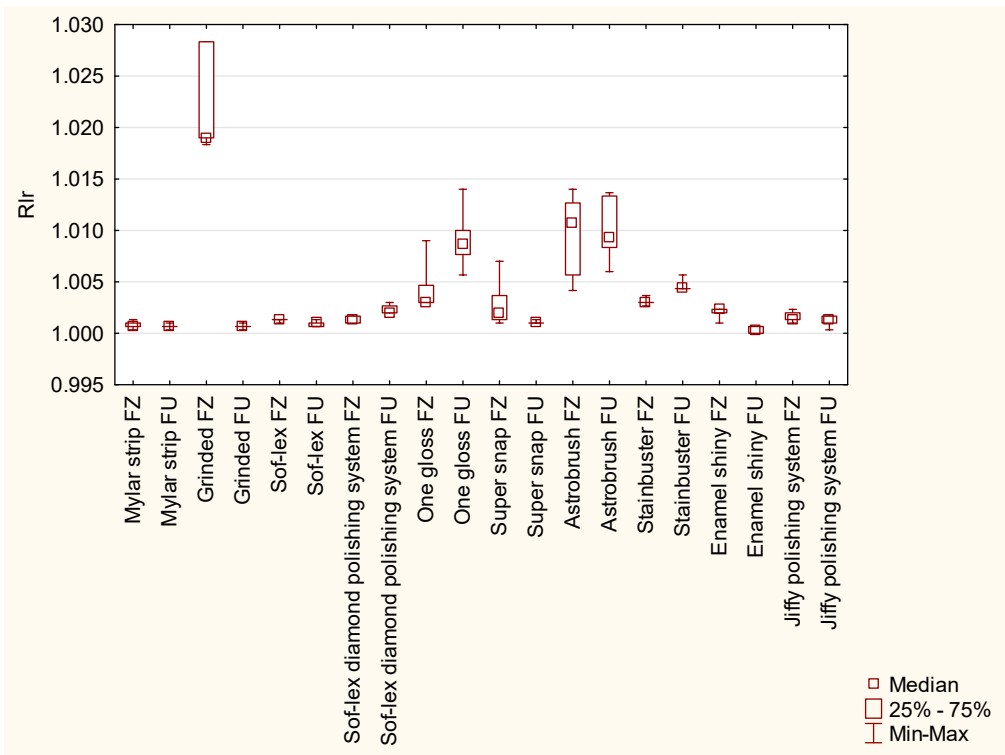

**Figure 2.** Median analysis, Rlr comparison between FU and FZ.

The highest Ra values after polishing are recorded for One Gloss for FU (0.657 μm) [Table 3, Figure 1], and Astrobrush for FZ (0.525 μm) [Table 4, Figure 1]. The highest Rlr coefficient is measured after polishing FU using Astrobrush and One Gloss (1.009) [Table 5, Figure 2]. For FZ, the highest Rlr parameter value is found after polishing with Astrobrush (1.011) [Table 6, Figure 2].

Among the one-stage systems, the most effective tool is Stainbuster, yielding values of Ra 0.470 μm and Rlr 1.004, for FU, and Ra 0.169 μm and Rlr 1.003 for FZ.

Statistically significant changes in the Ra parameter are noted after polishing FU with Sof-Lex and One Gloss ($p = 0.0121$; Kruskal–Wallis), and also Super Snap and One Gloss ($p = 0.0079$; Kruskal–Wallis) [Table 3]. For the same material, significant differences in Rlr are found between the efficiency of Enamel Shiny and One Gloss ($p = 0.0066$; Kruskal–Wallis), and Enamel Shiny and Astrobrush ($p = 0.0046$; Kruskal–Wallis) [Table 5].

For FZ, significant differences are observed in the Ra after polishing with Super Snap and One Gloss ($p = 0.0297$; Kruskal–Wallis), and with Super Snap and Astrobrush ($p = 0.0079$; Kruskal–Wallis) [Table 4]. No significant differences are found in Rlr between polishing systems.

No statistically significant differences are found between the smoothness of the materials based on the application of a specific polishing system. However, some trends are visible. After applying Sof-Lex, Super Snap, Enamel Shiny, and Jiffy Polishing System, a smoother surface is observed for FU, while after polishing with Sof-Lex Diamond Polishing System, Stainbuster, One Gloss, and Astrobrush, a smoother surface is observed for FZ, which is illustrated in Figures 1 and 2.

Sample photos of samples after binding under the Mylar strip, after grinding, and after polishing with systems that produce the best smoothness and the roughest surface are presented in Figures 3 and 4.

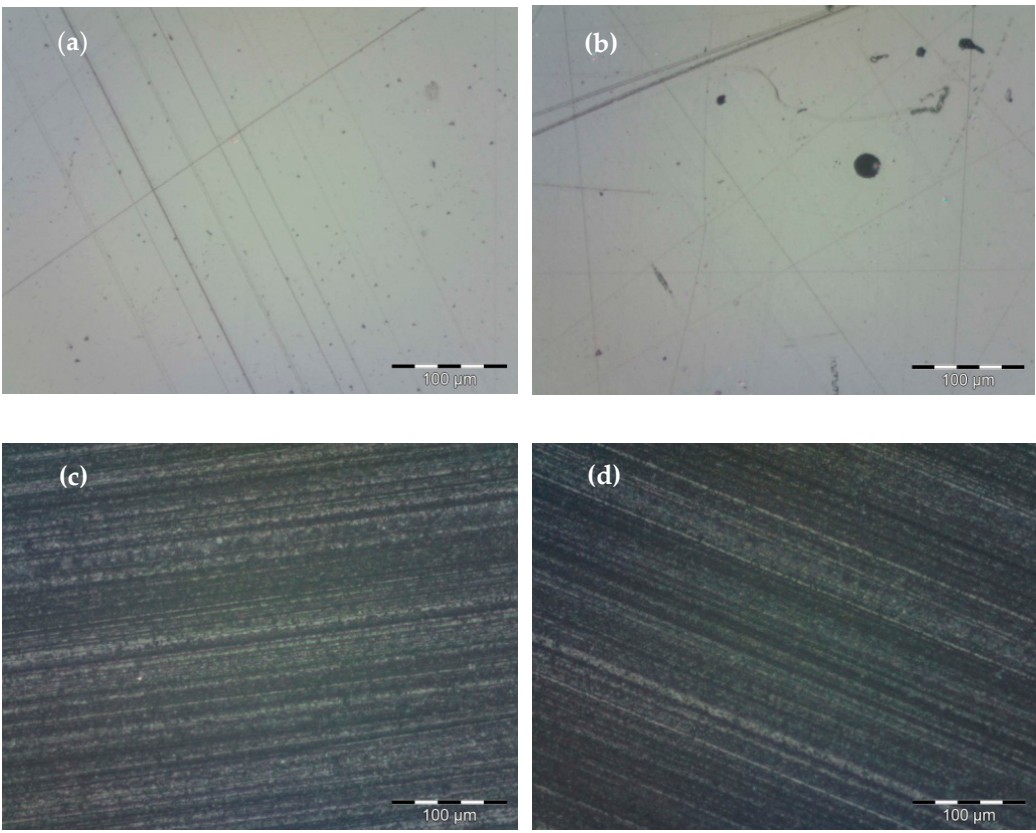

**Figure 3.** Photos from an optical microscope at 100 times magnification: (**a**) FU after binding under Mylar strip; (**b**) FZ after binding under Mylar strip; (**c**) FU after grinding with 600 grit sandpaper; (**d**) FZ after grinding with 600 grit sandpaper.

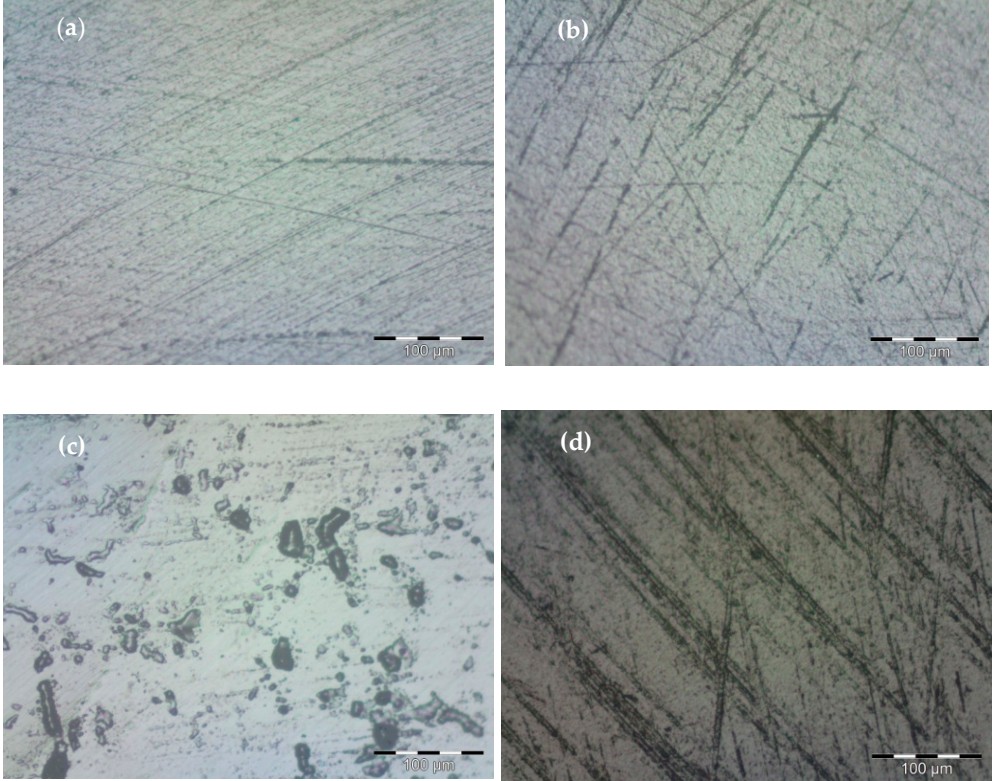

**Figure 4.** *Cont.*

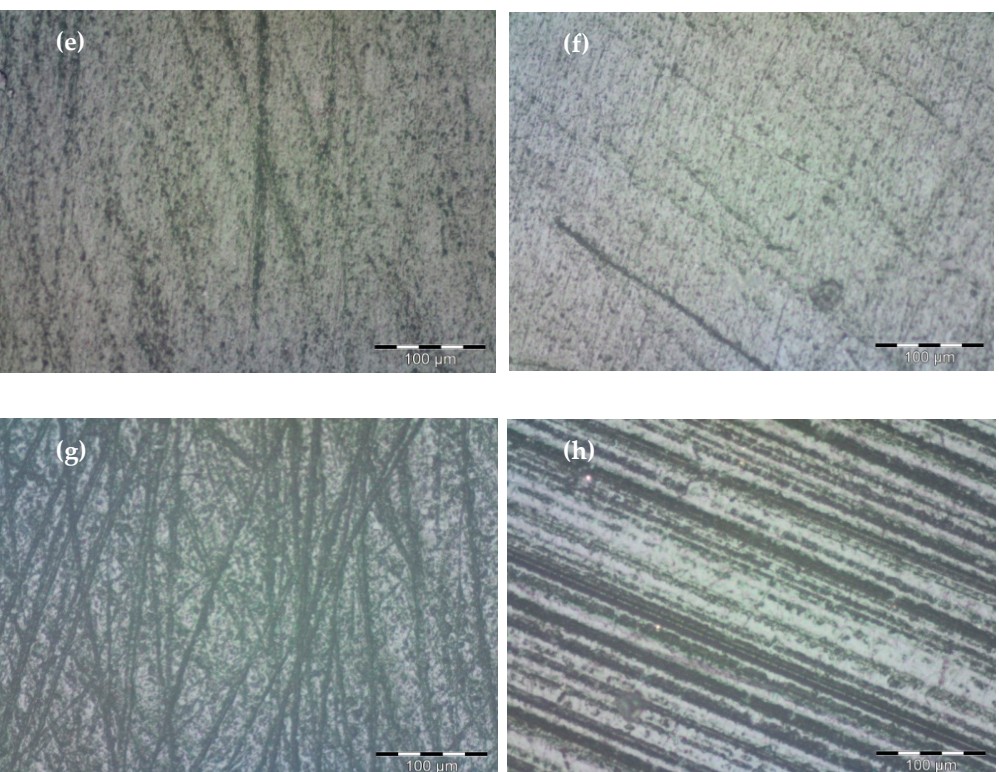

**Figure 4.** Photos from an optical microscope at 100 times magnification: (**a**) FU after polishing with Super Snap; (**b**) FZ after polishing with Super Snap; (**c**) FU after polishing with Enamel Shiny; (**d**) FZ after polishing with Sof-lex; (**e**) FZ after polishing with Sof-lex Diamond Polishing System; (**f**) FZ after polishing with Jiffy Polishing System; (**g**) FU after polishing with One Gloss; (**h**) FZ after polishing with Astrobrush.

## 4. Discussion

The materials selected for this study are very similar in resin and filler composition, but differ in the size of filler particles [Table 1]. Filtek Z250 is a microhybrid composite resin, with a filler particle size from 0.01 to 3.5 μm. The filler, which contains zirconium and silica particles, constitutes 60% of the volume. In Filtek Ultimate, nanoparticle material, the filler consists of silica particles with a diameter of 20 nm, and zirconia with a diameter of 4–11 nm. Some of them do not form agglomerates, and some are bunched into clusters. The filler content is 55.6% by volume.

The main advantage of hybrid composites is their combination of filler particles that allows the highest levels of filler loading among resin composites, and a corresponding improvement in physical properties [24]. According to some studies, microhybrids have high strength and wear resistance, and can be polished to a high degree [25,26].

However, the resulting polish of hybrid composites is not long lasting [27]. This is the main reason for the development of nanocomposites, which offer better polish and gloss retention compared to hybrid composites [28], as confirmed by numerous studies [29–31]. Besides the improved surface smoothness, nanocomposites presents reduce polymerization shrinkage [15], improve color stability and superior esthetics [20,32].

When polishing a composite material, two phenomena may occur that affect the final smoothness of the filling surface. Firstly, due to its lower hardness, the resin is selectively wiped off, which results in the protrusion of filler grains above the filling surface on a microscopic scale [33].

The second process is the selective extraction of individual filler particles from the mass of the material. As a result, there are holes in the material whose diameter corresponds to filler particles. The larger they are, the rougher the surface of the polished filling becomes.

The choice of materials was also based on the fact that Filtek Ultimate has not been tested for roughness after using polishing systems other than Super Snap. The effect of the application of the One Gloss tool and the Enamel Shiny system on the surface roughness of Filtek Z250 has also not been studied yet. There are also no reports in the literature on the effect of the application of the Stainbuster tool on the surface roughness of both tested composites (Filtek Ultimate and Filtek Z250).

The materials selected for the study have the most possible similar composition of both the organic (resin) and inorganic (filler) parts. In the conducted study, the most important difference is the size of the particles of the filler embedded in the resin, and this seems to have the greatest influence on the final smoothness of the materials

Traditionally, it is believed that the ability to polish composites varies depending on their particle size [34]. Materials with smaller particles generally have a high polishability [35]. A number of studies report that materials with only a nanofiller are better polished than hybrid materials [29,34,36,37]. However, this is not confirmed in the present study: it is not evident that the nanoparticle material is characterized by a significantly better smoothness after polishing compared to the microhybrid material, as expected. Despite this, some trends are noted that show that, in some of the systems, a better polishing effect is obtained for the microhybrid Filtek Z250 material, compared to the nanoparticle Filtek Ultimate material. After polishing with Sof-lex Diamond Polishing System, Stainbuster, and One Gloss, a smoother surface is observed for FZ. On the other hand, after polishing with Sof-lex and Enamel Shiny, FU demonstrates lower roughness than FZ.

This situation might have occurred due to the fact that, although FU contains filler particles with an objectively smaller diameter than FZ, these particles tend to clump into larger groups, thus, forming indivisible conglomerates. The diameter of the conglomerates can be a factor that determines the final smoothness of the filling. Therefore, the smoothness of the filling surface depends on the quantitative ratio of free particles with a diameter of nanometers (nano-sized filler particles) to the created agglomerates.

This would mean that the material polishability is influenced not only by the size of the filler particles, but also by their final spatial arrangement in the polymerized material.

It must also be kept in mind that the surface roughness of two composites with the same average filler size is also influenced by the size of the largest fillers, as this has a greater influence on surface roughness than the average particle size [34]. In the conducted research, this would be the size of clusters created in FU. Some of these might be larger than the largest particles of FZ filler. In order to verify this thesis, a microscope analysis seems to be promising. It could enable the assessment of the true size of the clusters and their number compared to non-agglomerated nanoparticles.

However the structure of a composite can affect the results of its polishing, as surface roughness is not an intrinsic property of a material. Some authors state that polishing systems play a more important role in producing a smoother surface of composites than the structure does itself [6].

The effect of the polishing system action on the material surface depends on its base material shape and flexibility, and also on the features of the embedded abrasive particles, such as their hardness, shape, and size [38].

For a finishing system to be rendered effective, the cutting particles should be harder than the filler particles [39]. Otherwise, the polishing agent will only remove the soft resin matrix, and leave the filler particles protruding from the surface [40]. This may result in greater surface roughness.

In this experiment, polishing systems were selected from different groups in terms of polishing material, tool construction, and the number of tools used as a sequence (single or multiple steps). The best results are obtained after polishing both composites with the Super Snap and Sof-Lex systems. Similarly, a study by Kemaloglu, Karacolak, and Turkun on eight different polishing systems finds Super Snap to yield the best results regarding the surface of the composite material [41], while Mukhija, Kandaswamy, and Venkatesh find Sof-lex to be the best of three polishing systems regarding resin composite roughness [15].

Super Snap and Sof-Lex are abrasive discs made of flexible plastics coated with aluminum oxide grains. Aluminum oxide has a hardness value of 9 Mohs, which may be the optimal hardness for evenly removing filler particles and resin matrix from the examined materials [15].

Polishing with Super Snap results in a smoother surface compared to Sof-Lex (both contain aluminum oxide in fine and superfine discs—Table 2). This may be attributed to the fact that Sof-Lex discs are less flexible than Super Snap discs, resulting in increased pressure while polishing. That causes a deeper penetration of abrasive particles into the material, resulting in deeper scratches.

The abrasive material used in the Sof-Lex Diamond Polishing System, Enamel Plus Shiny, and Jiffy Polishing System is diamond, with a hardness of 10 Mohs [15]. These polishing systems result in the roughest surface after polishing. It is possible that diamond particles produce deeper scratches on a polished composite, which results in grater roughness of the examined material.

The significant rigidity of the base material in One Gloss can also cause worse polishing results. This tool demonstrates less favorable efficiency in studies conducted by Patel, Chhabra, and Jain, who find it results in the highest material roughness of three tested polishing systems. The authors also suggest that the poor efficiency of the abrasive system may be related to the low flexibility of the backing material, in which the abrasive is embedded [33]. In turn, the greater roughness of materials noted after polishing with Astrobrush may be due to the construction of the tool itself; although it has stiff bristles, these might bend due to the centrifugal force of the rotor, resulting in bristles diverging, and leaving unpolished areas of the material.

The results obtained after polishing materials with the Stainbuster are promising, especially in the case of polishing the microhybrid material (FZ). It results in the third smoothest surface after polishing among the chosen tools and systems (Ra 0.169 μm). Unfortunately, due to the lack of information on the exact structure of the drill parts, it is difficult to assess which feature is decisive in the polishing process. The manufacturer does not provide the exact composition of the tool building material, and no studies evaluating the effectiveness of this tool have been found in the literature.

In the current study, the best smoothness of both tested materials were obtained by the Sof-Lex and Super Snap systems. Both systems result in Ra values below 0.2 μm, which is described by researchers as a condition to limit bacterial adhesion. However, although these systems produce the best filling smoothness, they are not free of disadvantages: they can only be applied on flat areas of the tooth, and cannot effectively be applied on chewing surfaces.

In the present study, the best results are achieved after the application of the system with the longest polishing time. However, in the authors' opinion, this is not the most contributing factor to the superior efficiency of Super Snap compared to the remaining polishers. Geiger, Ravchanukayev, and Liberman also find that only slight differences are noticed between surfaces polished for 10 and 30 s, indicating that duration of polishing is of small clinical relevance. [42]. It is worth noting that the importance of polishing time must depend on the steps used in the polishing protocol.

Certain limitations of this experiment should be considered. The research was carried out using two types of composite materials: nanoparticle and microhybrid. Therefore, the present results cannot be extrapolated to all restorative materials, such as other composite materials with a different type, size, and configuration of filler particles, or glass-ionomer cements. Additionally, the A2 shade of materials was used in the study, and the following polishing systems: Sof-lex, Sof-lex Diamond Polishing System, One Gloss, Super Snap, Astrobrush, Stainbuster, Enamel Shiny, and Jiffy Polishing System. Thus, other colors and the applications of different tools and polishing systems should be investigated. In the polishing process, changes in the surface layer of the material are observed as selective extraction of filler particles from the mass of material, or wiping the resin leaving protruding filler particles. Therefore, a comparison of the hardness and chemical composition before

and after polishing seems to be an important issue. Moreover, two roughness coefficients were applied to assess the surface roughness: Ra, often found in the literature, gives the opportunity to compare the obtained results, and Rlr, which allows for a more precise analysis of the surface roughness of the tested materials. However, it should be noted that, in order to obtain more comprehensive analysis of the roughness assessment, other coefficients should also be verified. The samples were polished with or without water spray, according to the manufacturer's recommendation, and the geometric structure of the material surface was assessed after 24 h. The long-term performance of polished materials under simulated conditions (thermocycling and various humidity conditions) would be greatly useful for developing a standard for polishing filler materials in clinical conditions. Additionally, further clinical studies investigating that area should be performed.

## 5. Conclusions

Within the limitations of this study, the following conclusions may be drawn:

1.  The use of different polishing systems results in varying surface roughness;
2.  The structure of composites does not significantly affect their surface roughness after polishing. Materials with different sizes of filler particles have similar polishability;
3.  In the presented study, the most effective polishing system is Super Snap.

**Author Contributions:** Conceptualization, M.E.S., M.K. and E.B.-R.; methodology, M.E.S. and M.K.; software, M.E.S. and M.K.; validation, M.E.S., M.K. and E.B.-R.; formal analysis, E.B.-R.; investigation, M.E.S. and M.K.; resources, M.E.S.; data curation, M.E.S. and M.K.; writing—original draft preparation, M.E.S.; writing—review and editing, E.B.-R.; visualization, M.E.S.; supervision, E.B.-R.; project administration, E.B.-R.; funding acquisition, M.E.S. and E.B.-R. All authors have read and agreed to the published version of the manuscript.

**Funding:** The research was supported by the Polish Ministry of Science and Higer Education and by Medical University in Łódź (grant no. 502-03/2-044-01/502-24-068) Poland.

**Institutional Review Board Statement:** Not applicable.

**Informed Consent Statement:** Not applicable.

**Data Availability Statement:** Not applicable.

**Conflicts of Interest:** The authors declare no conflict of interest. The funders had no role in the design of the study; in the collection, analyses, or interpretation of data; the writing of the manuscript, or the decision to publish the results.

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
