# Peer review of "The Effect of Various Polishing Systems on the Surface Roughness of Two Resin Composites—An In Vitro Study"

_coatings, doi:10.3390/coatings12070916_

Round 1

Reviewer 1 Report

Dear authors,

differences between both resins were expected as a result of differences in inorganic filler particles. As commented previously, the manuscript does not present any novelty or new information regarding the polishing of resin composites. Additionally, the sample size seems to be small hence non-parametric statistical test was used for comparisons. 

Author Response

Response to Reviewer 1 Comments

Thank you very much for your review.

After carefully examining the suggestions of the reviewers, the authors made corrections in the article.

The language corrections were made by a native English-speaking lecturer to improve the linguistic quality of the manuscript.

We clarified the results by adding some numerical data in the abstract and also highlighted the fact that polishing with Super Snap allowed to obtain the best results among all of the tested polishing systems despite it did not turn out to be the best polishing systems in terms of the analysis of all tested coefficients. Super Snap gave the lowest Ra values for both materials, and in the case of Rlr, it was the second best in terms of the results obtained and only slightly worse than the systems that performed best results with regard to Rlr coefficient. Based on these facts, the authors chose the Super Snap system as the most effective what was included in the conclusion.

Authors also added optical microscope images presented the surface structure of some materials after binding under the Mylar strip, after roughening with 600-grit sandpaper and after polishing to demonstrate the visual changes in the surface structure of tested materials.

The purpose of our research was not only to compare the polishing properties of materials, but most of all to evaluate the effectiveness of various polishing systems and tools. They have not been compared so far in such number and configuration. According to the literature, it was the first time that the effectiveness of the Stainbuster tool was assessed against other polishing systems.

Reviewer 2 Report

Dear authors,

thank you for the renewed possibility to review the manuscript. It has improved in some points. However, there are still major points that need to be adapted, especially regarding english language and statistical analysis.

Abstract:

L. 17 and L. 25: Nanoposite and nanophilic are not regular terms. Please check. 

In general, the language of the manuscript should be thoroughly checked. 

L. 23: As mentioned before, you should add some numerical data already in the abstract.

Mat Meth:

Table 2: What does "heavy pressure" or "light pressure" mean? Have you standardized the pressure some how?

Please do not use multiple fonts in one table. 

L. 149: What dos "Due to the inconsistency with the normal distribution for the analyzed variables" mean? If part of the data is not distributed normal you must use non-parametrical tests.

Discussion: L.336-339: The importance of polishing time must in the opinion of the reviewer depend on the steps used in the polishing protocol. Please add this to the discussion.

Author Response

Response to Reviewer 2 Comments

Point 1: Abstract L. 17 and L. 25: Nanoposite and nanophilic are not regular terms. Please check. 

Response 1: Instead of the word „nanophilic” we used the term „nanoparticle” which is commonly used in English-language articles. The word “nanoposite” was used as a mistake due to incorrect spelling and so we removed it from the manuscript.

Point 2: In general, the language of the manuscript should be thoroughly checked. 

Response 2: According to your suggestion the article was reviewed by a native English-speaking lecturer and major linguistic corrections were made. They are marked up in the text using the “Track Changes” function in red.

Point 3: Abstract L. 23: As mentioned before, you should add some numerical data already in the abstract.

Response 3: Authors added some numerical data in the abstract which is marked up in the text.

Point 4: Mat Meth: Table 2: What does "heavy pressure" or "light pressure" mean? Have you standardized the pressure some how?

Response 4: In the case of the One Gloss tool, the manufacturer declares that it is a single tool that allows to obtain optimal polishing results. He recommends polishing in two steps - with less pressure followed by more pressure. Light pressure” was considered as comparable to the pressure used during finishing an polishing with other tools and systems. Such a ratio was possible due to the fact that all samples were polished by a single operator. Then, an attempt was made to polish the sample stuck to the tray placed on the balance in order to measure the applied force. The authors assumed the “heavy pressure” as a pressure about twice as high, calculated on the basis of the measured weight value (the pressure force is the mass multiplied by the gravitational acceleration).

Point 5: Please do not use multiple fonts in one table. 

Response 5: We unified the fonts using in tables the same font as in the main text.

Point 6: L. 149: What dos "Due to the inconsistency with the normal distribution for the analyzed variables" mean? If part of the data is not distributed normal you must use non-parametrical tests.

Response 6: We clarified the information that part of the data was not distributed normal and that for that reason we used a non-parametrical test (Kruskal-Wallis test).

Point 7: Discussion: L.336-339: The importance of polishing time must in the opinion of the reviewer depend on the steps used in the polishing protocol. Please add this to the discussion.

Response 7: Authors added to the discussion that the importance of polishing time must depend on the steps used in the polishing protocol (the last sentence in section “Discussion”).

Reviewer 3 Report

Authors did improve the manuscript.

In introduction, the new section focusing on roughness is most welcome, as well as the description of the hypotheses of the study.

The added details in the materials & methods area improve the clarity of the paper.

Changes in the results, discussions and conclusion increase the readability of the manuscript, therefore I reccomend and accept of the article.

Author Response

Response to Reviewer 3 Comments

Thank you very much for the positive review. According to your suggestion, the conclusions have been revised. Authors highlighted in manuscript the fact that polishing with Super Snap allowed to obtain the best results among all of the tested polishing systems despite it did not turn out to be the best polishing systems in terms of the analysis of all tested coefficients. Super Snap gave the lowest Ra values for both materials, and in the case of Rlr, it was the second best in terms of the results obtained and only slightly worse than the systems that performed best results with regard to Rlr coefficient. Based on these facts, the authors chose the Super Snap system as the most effective.

Reviewer 4 Report

The topic "The Effect of Various Polishing Systems on the Surface Rough-2 ness of Two Resin Composites-In Vitro Study" is interesting. The manuscript was well written. However, the following specials should be confirmed.

(1)Filtek Ultimate (FU) 104 - nanocomposite and 40 of Filtek Z550 (Micro-hybrid)  are used.   Some tests should be done to confirm that the materials is nanocomposite or Micro-hybrid.

(2) The roughness on different surface sites is different. The author should give some AFM images or other test, or explain the surface area.

(3) Some SEM or optical images should be given.

Author Response

Response to Reviewer 4 Comments

Point 1: Filtek Ultimate (FU) 104 - nanocomposite and 40 of Filtek Z550 (Micro-hybrid)  are used.   Some tests should be done to confirm that the materials is nanocomposite or Micro-hybrid.

Response 1: Authors based on the data provided by the manufacturer. It is a well-known brand on the dental market, and the materials were tested and provided with appropriate approvals before being released to the market. For this reason, the authors adopted the specification provided by the manufacturer.

Point 2: The roughness on different surface sites is different. The author should give some AFM images or other test, or explain the surface area.

Due to the possibility of formation on polished materials paths depending on the direction of the operator's movements, the samples were subjected to profilometric measurement in the following way: on each sample, three separate measurements were made and during each of them the profilometer needle traveled a distance of 4 mm. Their directions were inclined to each other by an angle of 60 degrees. The value obtained in this way for each of the samples was averaged and these averages were subjected to statistical analysis. In the opinion of the authors, performing measurements three times on a given sample in different places in the given manner allowed for averaging the measurements in the context of discrepancies in the roughness of samples in different places on their surface.

Point 3: Some SEM or optical images should be given.

Response 3: As You rightly suggested authors added optical microscope images presented the surface structure of some materials after binding under the Mylar strip, after roughening with 600-grit sandpaper and after polishing to demonstrate the visual changes in the surface structure of tested materials.

Round 2

Reviewer 1 Report

Dear authors,

although this is an interesting topic, there is no novelty in this study. Stainbuster has been tested since 2008, with manuscripts published in Pubmed comparing this bur to another polishing protocols. Thus, I recommend the manuscript to be rejected.

Author Response

Thank you again for your review and for your time spend to read corrected manuscript.

Reviewer 2 Report

Dear authors,

thank you or the renewed possibility to review the manuscript.

Here are some few points that can be adapted.

L. 33: Although removed mostly, the confusing term "nanophilic" is still present in the abstract.

Table 2: Please use uniform font sizes in Table 2. Furthermore and as mentioned before, the brand 3M Espe does not exist anymore. PLease replace by "3M Oral Care" and add Company, City, Country to any material during the whole manuscript.

Table 2: In your rebutttal letter, you mentioned that you measured the weight representing heavy and light pressure during polishing. Please add the measured weight somewhere, favourably directly in the table 2.

Author Response

Response to Reviewer 2 Comments

Thank you again for your review. The authors referred to the suggested corrections.

Point 1: L. 33: Although removed mostly, the confusing term "nanophilic" is still present in the abstract.

Response 1: We replaced the word “nanophilic” with the word “nanoparticle as a corrected form. We apologize for our earlier oversight.

Point 2: Table 2: Please use uniform font sizes in Table 2. Furthermore and as mentioned before, the brand 3M Espe does not exist anymore. Please replace by "3M Oral Care" and add Company, City, Country to any material during the whole manuscript.

Response 2: The authors adhered to the above recommendations unifying the fonts in tables and correcting the information about the manufacturers (after contacting representatives of individual companies). We assumed that we report the manufacturer of the material or tool each time it appears for the first time in the manuscript. Should we add the manufacturers of polishing tools when they appear for the first time in the text or is it enough that the information is given in the table preceding their first appearance in the text? We would be grateful for any hints as to whether such a procedure is correct.

Point 3: Table 2: In your rebuttal letter, you mentioned that you measured the weight representing heavy and light pressure during polishing. Please add the measured weight somewhere, favourably directly in the table 2.

Response 3: The measured weight with repeated measurements was oscillated around value 30g for standard polishing. The operator (the corresponding author) practiced a pressure that oscillated around value 100 g for the “heavy pressure” polishing which was applied during the first step polishing with the One Gloss. After converting the values obtained on the balance to the pressure force, assuming that at a steady acceleration due to gravity the weight is proportional to the mass it gives values of about 1,0 N for the “heavy pressure” and about 0,3 N for the “light pressure” polishing. These values have been added to table 2. We also we also added the applied tip speed which was not included previously.

This manuscript is a resubmission of an earlier submission. The following is a list of the peer review reports and author responses from that submission.

Round 1

Reviewer 1 Report

Minor English-language related changes are necessary, such as the use of "." instead of "," as a decimal; "prosperity" in the abstract might be changed etc. I would suggest an extension of the conclusion section, in order to cover all the analysed parameters.

The variety of polishing systems is good. However, a more extensive comparison would be welcome.

Reviewer 2 Report

Dear authors, 

thank you for the opportunity to review your manuscript dealing with an interesting topic. However, there are some major critical points that contradict publication in the present form as listed below:

Abstract:

The statistical analyses used in this study should be namend in the abstract. Please name some of the numerical results in the abstract. 

Introduction:

L. 39: " but smaller by at least one order of magnitude. "

Please be more concise about the exact measurements, units and procedures. Parts of the introduction are too generalistic for a scientific article. 

L. 40: The deinition of Ra is imprecise. A formula or scheme would help. There are also some ISO-norms defining the exact procedure for surface roughness assessments. Please refer to these norms.

The same goes for Rlr later in the manuscript. 

paragraph 56-61: 

One benefit of a mylar strip polymerization could be, that there is no oxygen inhibition layer with these technique. If you polymerize without mylr strip or glycerine, you have to preparate the surface anyway with instruments. On the other hand, using a mylar strip highers the c-factor, which could be unfavourable for mechanical properties of the restorations. 

Maybe add these points here or to the discussion. 

L. 86 ff: The introduction lacks a sound null-hypothesis for the study at the end. 

M&M:

Table 1: The brand is called 3M Oral care since a couple of years, as ESPE is not longer existant. Please double-check, if you used older materials (then 3M ESPE would be correct) or adapt the brand name. 

L. 101: Please use the correct unit for light-power (mW/cm^2) and add information about the working distance during polymerization. Have you checked/measured the light-power of your lamp during the study?

L. 106: How long and with how much pressure have you roughened the surface with sandpaper? Please provide more information. 

Table 2: Have you somehow standardized the pressure during application?

Table 2: Why have you used different times for each polishing system? To the reader it is not suprsing that supersnap with the longest polishing time reveals the best results. As they may be used in different handpicks, it would be more valuable to use same revolution numbers for all systems. 

L. 119: What does "120o directional difference" mean? Maybe a scheme showing measurement systematics would be helpful for the readers. 

L. 131: As you decided to treat data non.parametrically (which is correct as they are not normally distributed), you should remove all standard deviations and provide medians and 25-75% quantiles instead. The mneand and SD values and graphs could be removed completely. 

Results: 

Figure 4 etc.:

In the manuscript, you used the abbreviation FZ, in the figures Z250. Please be consistent.

I am not a statistician, but I feel quite bad about the statistical analysis. I would suggest the following tests: Mann Whitney U pairwise comparison of every polishing procedure against the Mylar strip and pairwise. Furthermore, every FU group should be analysed against every FZ group. 

Discussion:

L. 231: To the understanding of the reviewer, the differences between the materials ar widely discussed here, but not investigated in the study by statistical analysis (Mann Whitney U tests between groups and Error Rates Method for the whole material). Please add, otherwise the discussion about the two materials is pointless. 

L.269 ff: Please discuss the point that supersnap was used longer than every other polishing system. 

Conslusion:

The discussion and conclusion need to be written clearer, as the study does not answer if there is an difference between FZ and FU and it only concludes that super snap led to the best results with the clear limitation of non-complex specimens and longest polshing duration for super snap.

Reviewer 3 Report

Dear authors,

The manuscript as presented should be sent to a dentistry journal focused on operative dentistry. There is no novelty regarding materials surface treatment/coating. Please, check the power of the statistical test hence Kruskal Wallis is indicated for non-parametric data.

Reviewer 4 Report

Dear authors,

The article does not present the novelty of the research. What does this research bring new compared to the already existing ones? This can be seen from the chosen bibliography. Many of the cited works are before 2015. Then the methodology is not clearly presented: how to polish, with what tool, then how to determine the roughness, what other characterization methods could you use? What was the purpose of the roughness investigations, and what is the application? SEM investigation could help to improve your manuscript. Is the chosen method good? How do I investigate if I chose the right materials? Then, a paragraph at the end is not enough to conclude the research!